# Highly Selective Electrochemical CO_2_ Reduction to C_2_ Products on a g-C_3_N_4_-Supported Copper-Based Catalyst

**DOI:** 10.3390/ijms232214381

**Published:** 2022-11-19

**Authors:** Zijun Yan, Tao Wu

**Affiliations:** 1New Materials Institute, University of Nottingham Ningbo China, Ningbo 315100, China; 2Key Laboratory of Carbonaceous Wastes Processing and Process Intensification of Zhejiang Province, University of Nottingham Ningbo China, Ningbo 315100, China; 3Nottingham Ningbo Beacons of Excellence Research and Innovation Institute, Ningbo 315000, China

**Keywords:** electrochemical reduction of CO_2_, graphite phase carbon nitride, copper oxide, C_2_ products, hydrothermal preparation for advantageous electrocatalyst

## Abstract

Herein, a novel approach used to enhance the conversion of electrochemical CO_2_ reduction (CO_2_R), as well as the capacity to produce C_2_ products, is reported. A copper oxide catalyst supported by graphite phase carbon nitride (CuO/g-C_3_N_4_) was prepared using a one-step hydrothermal method and exhibited a better performance than pure copper oxide nanosheets (CuO NSs) and spherical copper oxide particles (CuO SPs). The Faradaic efficiency reached 64.7% for all the C_2_ products, specifically 37.0% for C_2_H_4_, with a good durability at −1.0 V vs. RHE. The results suggest that the interaction between CuO and the two-dimensional g-C_3_N_4_ planes promoted CO_2_ adsorption, its activation and C-C coupling. This work offers a practical method that can be used to enhance the activity of electrochemical CO_2_R and the selectivity of C_2_ products through synergistic effects.

## 1. Introduction

According to statistics, the level of CO_2_ in the air is rising, which has led to global warming [1]. Compared with other measures, the chemical transformation of CO_2_ into carbon-containing chemicals is a promising option for CO_2_ mitigation. Among these options, electrocatalysis is considered to have high potential because it uses clean energy generated by electrical power and reacts in an aqueous solution at room temperature and pressure under mild reducing conditions [2,3]. As a result, it is environmentally friendly and transforms CO_2_ into fuels and chemical materials with significant added value [4].

Among the numerous metal catalysts, copper is the only one that has a moderate binding energy to intermediates, which can generate numerous C_1_ and high-value C_2_ and C_2+_ products [5,6]. Compared to copper, as a simple substance, oxide-derived copper catalysts at lower potentials exhibit a considerably enhanced CO_2_ electroreduction to C_2_ [7]. However, in conventional thermal conversion, it is difficult to meet the need for the increased reaction rates and selectivity of the target products at present. Although several novel synthetic strategies for copper-based catalysts have been reported, such as electrodeposition and plasma treatments [7,8,9,10], these complex synthesis methods often require the adoption of harsh reaction conditions and the use of expensive equipment, which hinder the widespread application of the related technologies. Consequently, it is necessary to create new methods that are simple and environmentally friendly for the preparation of catalysts.

Thus far, a wide range of CO_2_R techniques for oxide-derived copper catalysts have been reported. Ager et al. [11] prepared oxide-derived Cu catalysts based on Cu_2_O through electroreduction and obtained a selectivity of the C_2+_ products (FE = 60%) for at least 5 h at −1.0 V in 0.1 M potassium bicarbonate. Cui et al. [12] introduced N into Cu_2_O to produce nitrogen-doped Cu_2_O, which resulted in the enhanced CO_2_ adsorption and doubled Faradaic efficiency of ethylene (10%) compared to Cu_2_O. However, despite the excellent electrocatalytic performance of copper metal in regard to CO_2_, it is still affected by its complex reduction products and poor single-product selectivity and CO_2_R activity. Moreover, CO_2_ is poorly soluble in water. Thus, the overpotentials related to C_2_ generation reactions are high, which makes it more difficult for CO_2_ to be adsorbed on the catalyst surface and renders the competing hydrogen precipitation side reactions significant [13,14]. In this regard, Luo et al. [15] proposed that high CO_2_ pressure at the same concentration of the electrolyte can lead to a lower local pH, which will increase the surface coverage of CO, thus promoting the formation of C_2_ products. Wang et al. [16] created a Cu/N_x_C (nitrogen-doped carbon) interface and found that CO_2_ has a strong interaction with the Cu/N_x_C interface and is enriched on N_x_C, which increases the selectivity of C_2_ by 200–300%. Hence, there is a need to develop a novel catalyst that is more efficient and highly selective, following this line of reasoning.

Density functional theory (DFT) indicates that, for multi-carbon products, the ability of multiple *CO intermediates to engage in a C-C coupling process and influence the strength of *CO intermediate binding located in the catalytic active center are the determining factors [17]. In order to promote such an interaction, a number of enhancement studies have been undertaken, including the study of g-C_3_N_4_. It has a high thermal and chemical stability and special laminar structure, as well as a low cost, and its heterostructure provides a high CO_2_ adsorption and activation efficiency [18,19]. g-C_3_N_4_ contains a large amount of pyridine N. Theoretical calculations indicate that, as the main active site for electrocatalytic reactions, pyridine N, as a substrate, can complex with the metal nanoparticles to stabilize them and also provides an active center for CO_2_R. The interaction of g-C_3_N_4_ with metal also causes the metal surface to be highly electron-rich, thus enhancing the adsorption of reaction intermediates [20]. However, few experiments have been conducted to systematically verify this process. Jiao et al. [21] synthesized Cu-C_3_N_4_ to provide experimental evidence for the calculation of the g-C_3_N_4_ scaffold. In the synthesized samples, the analysis based on the N K-edge NEXAFS spectra and Cu 2p XPS confirmed the significant chemical interaction between N and Cu atoms. However, the main product of CO_2_R of Cu-C_3_N_4_ was still hydrogen (>50%). This suggests that the g-C_3_N_4_-loaded Cu-based catalysts still require further improvement. To date, there is no research that has been carried out to capitalize on the synergistic effects of g-C_3_N_4_ and copper oxide to enhance electrochemical CO_2_R and its selectivity.

In this work, starting with the material structure of electrocatalysts, CuO/g-C_3_N_4_ was prepared based on copper oxide catalysts using g-C_3_N_4_ as a carrier by a straightforward hydrothermal method combined with calcination. The morphology and components were analyzed, and the electrochemical performance and catalytic activity were investigated by drop coating the catalysts on carbon paper. During the discussion, the control experiments were performed using CuO NSs and CuO SPs, and the reasons for the efficient catalytic reduction to C_2_ products by the CuO/g-C_3_N_4_ electrode were comprehensively analyzed.

## 2. Results and Discussion

### 2.1. Electrochemical Activity Tests

LSV curves of the CuO SP, CuO NS and CuO/g-C_3_N_4_ catalysts in Ar- and CO_2_-saturated 0.1 M KHCO_3_ solution are displayed in Figure 1a. In the CO_2_-saturated electrolyte, clearly, all of the investigated catalysts showed higher current densities compared to those in the Ar-saturated electrolyte, demonstrating their great inherent activity for electrochemical CO_2_R. Under the same solution conditions, CuO/g-C_3_N_4_ also exhibited a smaller onset potential than the CuO NSs and CuO SPs. Additionally, CuO/g-C_3_N_4_ exhibited a noticeably improved current density from onset potential to −1.3 V vs. RHE relative to the other two CuO catalysts, thus implying its higher CO_2_R performance. Tafel curves for the overpotential-log (C_2_ current density) are plotted in Figure 1b. Compared with the CuO NSs and CuO SPs (27.8 mV·dec^−1^ and 33.6 mV·dec^−1^), the slope of CuO/g-C_3_N_4_ exhibits a significant decrease (17.2 mV·dec^−1^), which provides the further evidence of the better intrinsic properties of the CuO/g-C_3_N_4_ surface. The advantageous current density is also attributed to the lower charge transfer resistance of CuO/g-C_3_N_4_, reflecting the improved charge transfer process at the interface of the electrode and surrounding electrolyte, according to the EIS measurements and the fitting results in Appendix A.

The ECSA of the CuO SP, CuO NS and CuO/g-C_3_N_4_ electrodes were studied by contrast with the corresponding C_dl_ (double-layer capacitance). The CVs of these three electrodes are presented in Appendix A. The slope of the non-faradaic capacitive current (current density at OCP) versus the scan rate was used to calculate the C_dl_ value. In Figure 1c, it is clear that the C_dl_ of CuO/g-C_3_N_4_ (3.6 mF·cm^−2^) is much higher than that of the CuO NSs (2.3 mF·cm^−2^), indicating that CuO/g-C_3_N_4_ has a larger electrochemical active surface area and more exposed active sites, which are advantageous for boosting the CO_2_R activity. This also proves that the larger active surface area is caused by the g-C_3_N_4_ layer. Meanwhile, the C_dl_ of the CuO NSs is larger than that of the CuO SPs (1.7 mF·cm^−2^), suggesting that the nanosheets of CuO have a larger active surface area than the spherical nanoparticles of CuO.

### 2.2. Electrochemical CO_2_ Reduction Performance Tests

Furthermore, the CO_2_RR gaseous product distributions of the CuO SPs, CuO NSs and CuO/g-C_3_N_4_ were comparatively studied using a potential region of −0.8~−1.2 V vs. RHE. As given in Figure 2a, the CuO SPs consistently preferred CH_4_ production in all the testing potential regions, while the C_2_H_4_ selectivity was very low (<20%). This result indicates that regardless of the applied potentials, CuO SPs do not exhibit a particularly high C-C coupling activity. Comparatively to the CuO SPs, whose Faradaic efficiency was not higher than 5% at −0.8~−1.2 V, that of CH_4_ was suppressed for the CuO NSs. At the best applied potential of −1.0 V, the C_2_H_4_ production increased from 16.5% to 31.7%, and some ethane was produced as well (Figure 2b). However, in terms of the CuO/g-C_3_N_4_ electrocatalyst, the CH_4_ and CO productions were considerably further suppressed, totaling less than 6% at the potential of −1.0 V, as Figure 2c shows, while the CuO NS catalyst had an 8.1% Faradaic efficiency of C_1_ gaseous products at the same potential. Meanwhile, the C_2_H_4_ selectivity in CuO/g-C_3_N_4_ was significantly improved, whereas the H_2_ production caused by the HER side reaction clearly decreased. In particular, at the potential of −1.0 V, the C_2_H_4_ Faradaic efficiency reached as high as 37.0%, which was accompanied with H_2_ formation at 25.8%. A very small amount of HCOOH (8.8%) was also detected at this potential in the liquid products. Ethanol was reliably detected as well. At −0.8~−1.0 V, its Faradaic efficiency varied in the small range of 27.3~28.2%. The Faradaic efficiency for C_2_ was 64.7% for CuO/g-C_3_N_4_ at the respective optimal potential for C_2_ electroproduction. In brief, the CuO/g-C_3_N_4_ catalyst achieves a further improvement in the selectivity of C_2_ products.

Appendix A compares the geometric partial current densities of several products for various electrodes. At each potential, the H_2_ formation rate for the electrodes follows the trend CuO/g-C_3_N_4_ <  Cu NSs  <  CuO SPs, illustrating the H_2_ production caused by the competing HER was well controlled on CuO/g-C_3_N_4_. Meanwhile, the maximum C_2_H_4_ and C_2_ partial current densities occurred at −1.0 V in CuO/g-C_3_N_4_, reaching 14.0 mA·cm^−2^ and 24.5 mA cm^−2^, respectively. The above results show the exceptional performance of CO_2_R compared to that of the previously reported Cu-based electrocatalysts (Appendix A).

CuO/g-C_3_N_4_ has an excellent activity and selectivity in addition to a good stability. Figure 2d shows its chronoamperometric responses after being biased for two hours at −1.0 V (corresponding to the highest C_2_H_4_ selectivity, especially for CuO/g-C_3_N_4_). The CuO/g-C_3_N_4_ and Cu NSs displayed an excellent stability in these two hours. In terms of CuO/g-C_3_N_4_, the total cathodic current density tended to smooth out and was still maintained at 37.0 mA·cm^−2^ at the end of the stability test. Its corresponding C_2_H_4_ selectivity experienced an initial increase and was then retained at over 37.0% throughout the electroreduction. The CuO SPs catalyst, on the other hand, demonstrated a quick catalytic deactivation. The current density dropped rapidly by 40% after 5000s, and there was also a downward trend in the Faradaic efficiency of C_2_H_4_ at the end, despite the fact that it experienced a spike during the previous experiment.

### 2.3. Structure and Morphology Characterizations

The surface characteristics of nano-adsorbents can be identified using N_2_ adsorption/desorption measurements. In Figure 3, for the CuO/g-C_3_N_4_ material, there is a mesoporous structure, as evidenced by the large hysteresis loop from 0.47 to 1.00 of P/P_0_, which can also be proved by its pore size distribution using the BJH method. Remarkably, the adsorbed quantity of CuO/g-C_3_N_4_ is more than that of the CuO NSs, demonstrating that CuO/g-C_3_N_4_ has a more consistent pore structure and a greater surface area [22,23]. The BET surface areas of the CuO NSs and CuO/g-C_3_N_4_, as obtained, are 7.62 m^2^/g and 11.2 m^2^/g, respectively. Meanwhile, a higher average pore volume of 0.048 cm^3^·g^−1^ is observed in CuO/g-C_3_N_4_ (versus 0.029 cm^3^·g^−1^ of the CuO NSs sample). In a word, g-C_3_N_4_ contributes to the increase in the surface area as well as the pore volume. Thus, CO_2_ can be better adsorbed, combined with the synergistic effects between CuO and g-C_3_N_4_.

The crystal phase of the synthesized catalysts was revealed by the XRD patterns. According to Figure 4a, the graphitic materials with two specific diffraction planes, (100) and (002), correspond to the two typical diffraction peaks of pure g-C_3_N_4_ at 13.08° and 27.17°, which are caused by the interlayer stacking of conjugated aromatic rings and the in-plane structure of tri-s-triazine motifs [24,25]. In terms of the CuO NSs, the main peaks at 32.50°, 35.50°, 38.73°, 38.96° and 48.73° can be attributed to the crystal facets (−110), (002), (111), (200) and (−202) of CuO (JCPDS#45-0937), some of which are also depicted in the HRTEM and SAED patterns (Appendix A). The CuO/g-C_3_N_4_ composite allows for the observation of both g-C_3_N_4_ and CuO XRD diffraction peaks, and the absence of any additional distinctive peaks indicates the high purity of the samples immediately after preparation. Meanwhile, the cluster bands between 1248 and 1631 cm^−1^ in the FTIR spectra of g-C_3_N_4_ (Figure 4b) can be classified as the classic stretching mode of C-N heterocycles, and the heptazine ring system is the source of the 805 cm^−1^ sharp peak. The peaks of Cu-O stretching vibrations are also very obvious. The distinctive bands seen in CuO NSs at 425 cm^−1^ correspond to the CuO Au mode, and those at the slightly higher wavenumber position of 496 cm^−1^ correspond to the CuO Bu mode [26,27,28,29]. All of them can be seen in CuO/g-C_3_N_4_. This evidence prove that the two elements, g-C_3_N_4_ and CuO, coexist in the CuO/g-C_3_N_4_ material.

The elemental compositions of g-C_3_N_4_ and CuO/g-C_3_N_4_ were examined by XPS spectroscopy, and the binding energies obtained for each of them were compared. The investigated spectra in Figure 5a confirm that all the anticipated elements are present, namely N 1s and C 1s for g-C_3_N_4_ and O 1s, Cu 2p, N 1s and C 1s for CuO/g-C_3_N_4_. In Figure 5b, the Cu 2p high-resolution peak of the composite CuO/g-C_3_N_4_ is observed. The pattern of CuO/g-C_3_N_4_ presents with a pair of peaks discovered at 932.7 and 952.5 eV that are associated with two typical energy levels of copper: 2p_3/2_ and 2p_1/2_. CuO crystals are present, according to the nearly 20 eV spin-orbit energy difference. This also indicates that the Cu in the sample is in the +II oxidation state. The small peaks at 940~945 eV are satellite peaks, which are generally seen in Cu ions in the oxidation state of +II. Furthermore, in Figure 5c, a C 1s binding energy peak is shown at 284.8 eV, which refers to g-C_3_N_4_ and CuO/g-C_3_N_4_, belonging to the C=N sp^2^ bond and the interaction of the metal oxide with g-C_3_N_4_ in the mixture. Similarly, on the s-triazine ring of graphitic nitride of g-C_3_N_4_ and CuO/g-C_3_N_4_, there are sp^2^ N-C=N bonds, which can be represented by the C 1s peaks at 288.1 and 287.9 eV. Assigned to pyridine nitrogen, pyrrolic nitrogen and graphitic nitrogen, respectively, the N 1s peak of g-C_3_N_4_ can be deconvoluted into three chemical states, i.e., the peaks at 398.5, 399.0 and 400.6 eV. In terms of pyridine N, the binding energy of the CuO/g-C_3_N_4_ sample (398.3 eV) is 0.2 eV lower than that of pure g-C_3_N_4_ (398.5 eV), impacting the interaction between the two molecules at the interface (Figure 5d) [30,31,32,33,34,35].

The morphology of g-C_3_N_4_, CuO and CuO/g-C_3_N_4_ were confirmed using FESEM (Figure 6) and TEM (Appendix A). In Figure 6a, the pure g-C_3_N_4_ is composed of irregular and loose aggregates of sheet-like structures. The lamellae are formed as a result of the thermal breakdown of the urea fracture while producing a large number of pores, and this rough appearance confers a very high functionalization on the g-C_3_N_4_ sheets. In Figure 6b, it can be seen that a large number of CuO nanosheets of varying lengths were synthesized, with an average width of about 300 nm. Figure 6c demonstrates that, in the case of the CuO/g-C_3_N_4_ composites, it is evident that the particles with smooth surfaces in the connected sheets of agglomerates are considered as CuO NSs loaded onto the g-C_3_N_4_ matrix. Meanwhile, the corresponding elemental mapping (Figure 6d–g) also shows that the C, N, O and Cu atoms are evenly distributed throughout the composite, indicating the emergence of the CuO/g-C_3_N_4_ structure and the close contact between them.

## 3. Materials and Methods

### 3.1. Materials

Urea, polyvinyl pyrrolidone K30 (PVP-K30), Cu(NO_3_)_2_·3H_2_O, sodium acetate (C_2_H_3_NaO_2_), NaOH, KHCO_3_, isopropanol (C_3_H_8_O) and acetone (C_3_H_6_O) were purchased from Sinopharm Chemical Reagent. Sodium dodecyl sulfate (SDS), ethanol (99.7%), Nafion-117 solution (~5%) and CuO SPs (40 nm, 99.5%) were purchased from Macklin. Carbon paper (GDS180S) was purchased from Ce Tech. Each chemical was used directly as received, and UP water (>18.2 × 10^6^ Ω·cm) was employed to dispense the whole aqueous solution.

### 3.2. Preparation of Catalysts

#### 3.2.1. Preparation of g-C_3_N_4_

A total of 20 g urea was placed into a 50 mL crucible, and after that, the prepared sample was placed in a muffle furnace, directly heated for 2 h in air conditions at 540 °C and a rate of 10 °C min^−1^. After allowing it to naturally drop to an ambient temperature, the g-C_3_N_4_ sample was obtained.

#### 3.2.2. Preparation of CuO NSs

A total of 0.238 g Cu(NO_3_)_2_·3H_2_O, 0.173 g C_2_H_3_NaO_2_, 0.500 g PVP-K30 and 0.295 g SDS were dissolved in 100 mL H_2_O. Then, under continuous rapid stirring, 0.1 M NaOH was added dropwise until a pH > 12 was obtained. The suspension was filled in a 200 mL stainless-steel autoclave with a Teflon lining and heated for 24 h at 170 °C. The resultant sample was obtained by centrifuging the suspension, repeatedly washing it in H_2_O and EtOH, and then heating it to 550 °C (5 °C min^−1^) and holding it there for two hours in air.

#### 3.2.3. Preparation of g-C_3_N_4_

The procedure was similar to that of CuO NSs, except for the fact that 0.100 g g-C_3_N_4_ was introduced in the first instance.

### 3.3. Characterization

The morphology and structure of the studied samples were characterized by field emission scanning electron microscopy (FESEM, Regulus 8100, 5 kV) and transmission electron microscopy (TEM, Tecnai G2 F20, 200 kV). Energy-dispersive X-ray spectroscopy (EDS) mapping was conducted using FESEM Regulus 8100. The atomic valence states and some molecular structures were investigated by X-ray photoelectron spectroscopy (XPS, Thermo Escalab 250), and the source gun type was Al Kα. X-ray diffraction (XRD) was carried out using D8 Advance, produced by German Bruker-AXS, operating at 40 kV and 40 mA with an accuracy of 0.01° (2θ) at room temperature. Fourier transform infrared spectrometer (FTIR) spectra were measured in the 400–4000 cm^−1^ range using a Nicolet iS50 FT-IR spectrometer, with the samples prepared as KBr pellets. Brunauer–Emmett–Teller (BET) surface area measurements were performed at 77 K using a TriStar II 3020 adsorption analyzer in the N_2_ adsorption mode.

### 3.4. Electrochemical Measurements

The carbon paper was pre-treated with acetone and washed at least 3 times with H_2_O and EtOH before being air-dried. A total of 4 mg of the sample (CuO/g-C_3_N_4_, CuO NSs or CuO SPs) was dispersed in 1 mL isopropanol and 30 μL Nafion solution, followed by sonication for 30 min to create the sample ink. Then, the ink was homogeneously added drop by drop onto the carbon paper (2 × 2 cm^2^) and dried on a hot plate, and then it was divided into working electrodes with a surface area of 1 × 1 cm^2^.

The electrochemical measurements were performed using an electrochemistry workstation (Gamry Reference 300) in a three-electrode system. A Nafion-117 membrane divided the two compartments of the H-cell. As the counter electrode and reference electrode, respectively, platinum grid and Ag/AgCl electrode (saturated KCl) were employed. CO_2_-saturated (pH ≈ 6.8) or Ar-saturated 0.1 M KHCO_3_ (pH ≈ 8.3) was used as the electrolyte. Additionally, each measurement potential was standardized to the reversible hydrogen electrode (RHE) reference scale, together with manual internal resistance compensation:(1)$$E_{RHE}=E_{Ag/AgCl}+0.059\times pH+0.197-iR_{u}$$

The *iR_u_* was determined by potentiostatic electrochemical impedance spectroscopy measurements under an open circuit potential (OCP) at frequencies ranging from 10^5^ Hz to 0.1 Hz. The linear sweep voltammetry (LSV) was tested in the same environment at a sweep rate of 10 mV·s^−1^. The current density equals the testing current divided by the geometric surface area of working electrode. The double-layer capacitance method was used to conduct the electrochemical active surface area (ECSA) tests. The potential range was OCP ± 50 mV, and cyclic voltammetry was performed at different sweep speeds. Gas chromatography (Agilent GC 8890) was used to identify the resulting gas phase products, and NMR (AVANCE III HD 400 MHz) was used to detect the products in the liquid phase.

### 3.5. Calculation of the Faradaic Efficiency

The following equation was used to calculate the Faradaic efficiency of the gas products:(2)$$FE_{gas}=\frac{nCGP\times 96485}{IRT}$$

Above, *n* is the amount of e^−^ that is transferred to the product formation. *C* is the concentration (ppm) of the gases revealed by GC. *G* is rate of CO_2_. *I* is the cell current. *P* = 1.01×10^5^ Pa. *R* is the universal gas constant. *T* = 273.15 K.

The Faradaic efficiency of the liquid products was calculated by following equation:(3)$$FE_{liquid}=\frac{cV\cdot ne\times 6.022\times 10^{23}}{\frac{Q}{e}}$$

Above, *c* is the concentration of the product. *V* is the total volume of the cathodic electrolyte and *e* is the electron. *Q* is the number of the transfer charge.

## 4. Conclusions

In summary, a CuO/g-C_3_N_4_ catalyst was fabricated by a simple hydrothermal method and achieved highly active and selective electrochemical CO_2_R to C_2_ products. The catalyst demonstrates a significant advantage over pure CuO nanosheets and spherical CuO particles and shows a high Faradaic efficiency of 37.0% for C_2_H_4_ at −1.0 V vs. RHE. It also has an ethylene catalytic stability that lasts for at least two hours. Meanwhile, the Faradaic efficiencies of all the C_2_ products of the composite are 64.7%, performing better than many other Cu-based catalysts, which indicates a synergistic promotion of C-C coupling between CuO and g-C_3_N_4_. Moreover, the structure and morphology characterizations demonstrated that the composite is based on g-C_3_N_4_-supported uniform polycrystalline copper oxide. The introduction of g-C_3_N_4_ increases the specific surface area, which promotes the mass transfer kinetics and provides new opportunities for the adsorption of CO_2_ and the exposure of active sites. Additionally, the interaction of pyridine N with copper oxide was confirmed, which further increases the reaction activity of CO_2_ reduction. This work provides an effective strategy that can be used to improve the selectivity and activity of C_2_ formation during the electrochemical reduction of CO_2_ and bridges the gap between the laboratory-based conversion of CO_2_ to economically valuable chemicals and its industrial application.

## Figures and Tables

**Figure 1 ijms-23-14381-f001:**
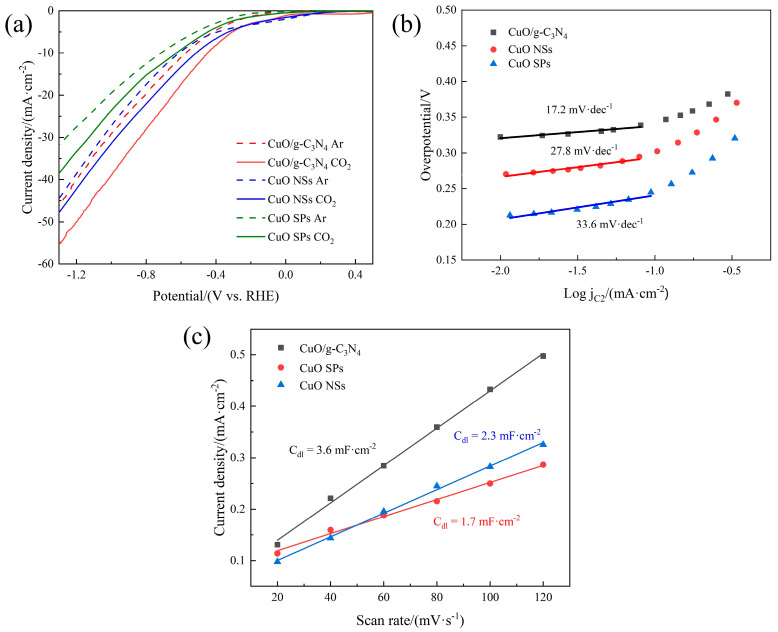
(**a**) LSV curves of CuO SPs, CuO NSs and CuO/g-C_3_N_4_ in Ar- and CO_2_-saturated 0.1 M KHCO_3_ solution, (**b**) Tafel slope calculation curves of CuO SPs, CuO NSs and CuO/g-C_3_N_4_, and (**c**) capacitive current at OCP as a function of the scan rate of CuO SPs, CuO NSs and CuO/g-C_3_N_4_ in CO_2_-saturated KHCO_3_ solution.

**Figure 2 ijms-23-14381-f002:**
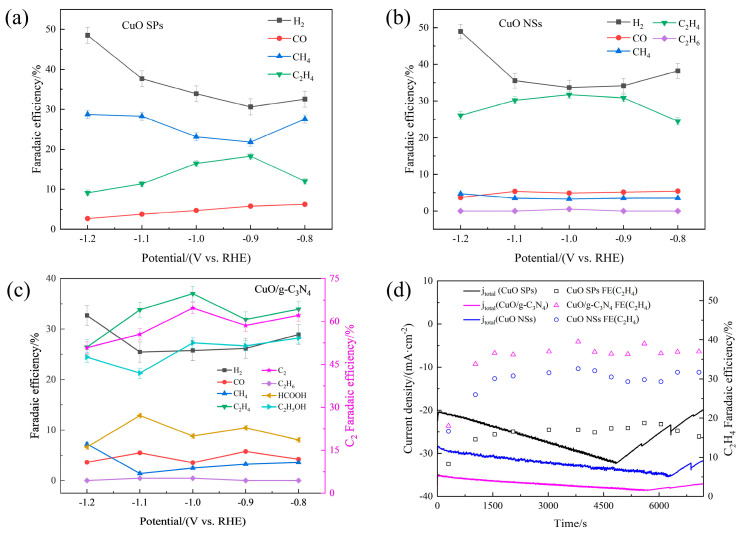
Faradaic efficiencies of CO_2_ electroreduction products as a function of the potential: (**a**) CuO SPs, (**b**) CuO NSs and (**c**) CuO/g-C_3_N_4_. (**d**) Stability tests for CuO SPs, CuO NSs and CuO/g-C_3_N_4_ at −1.0 V vs. RHE.

**Figure 3 ijms-23-14381-f003:**
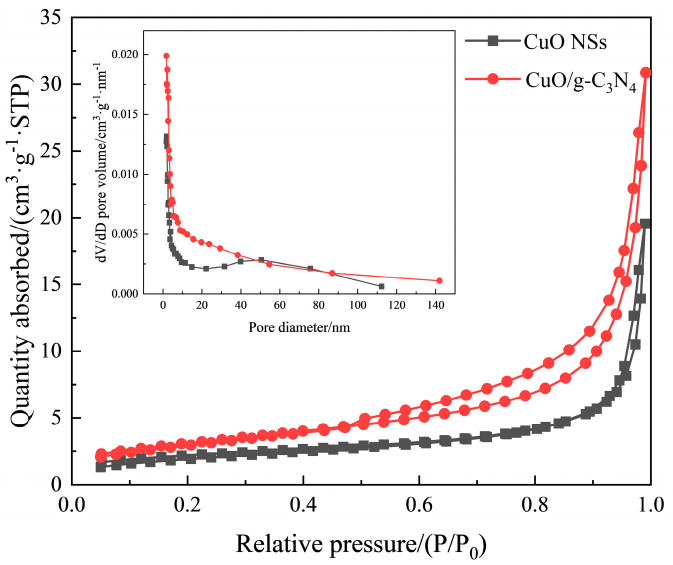
N_2_ adsorption–desorption isotherm and BJH pore size distribution plots (insets) of CuO NSs and CuO/g-C_3_N_4_.

**Figure 4 ijms-23-14381-f004:**
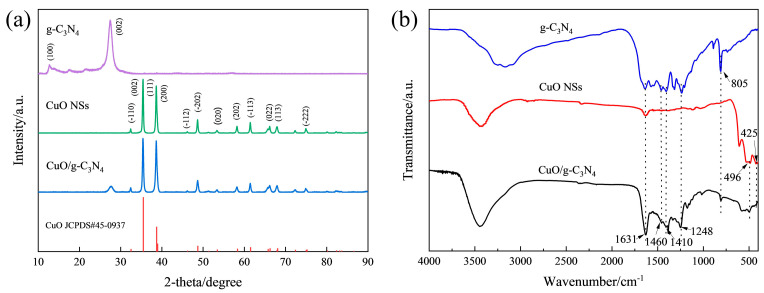
(**a**) XRD patterns of g-C_3_N_4_, CuO NSs and CuO/g-C_3_N_4_, (**b**) FTIR spectra of g-C_3_N_4_, CuO NSs and CuO/g-C_3_N_4_.

**Figure 5 ijms-23-14381-f005:**
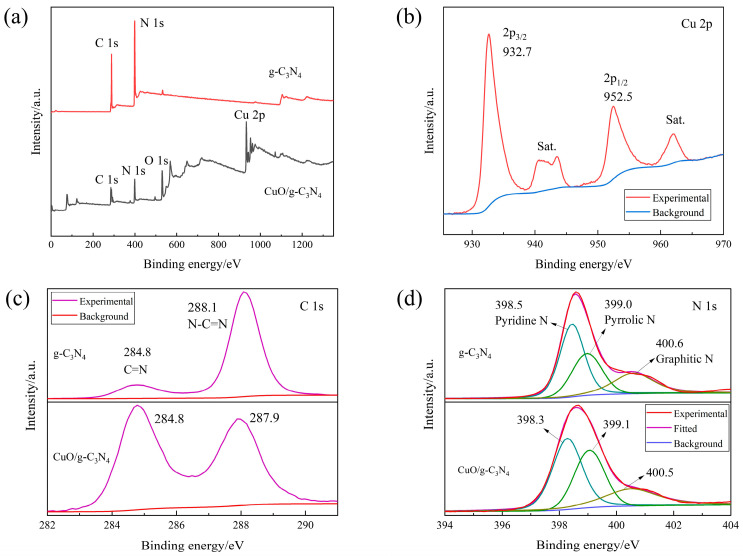
XPS spectra of g-C_3_N_4_ and CuO/g-C_3_N_4_: (**a**) full scan, (**b**) Cu 2p, (**c**) C 1s, (**d**) N 1s.

**Figure 6 ijms-23-14381-f006:**
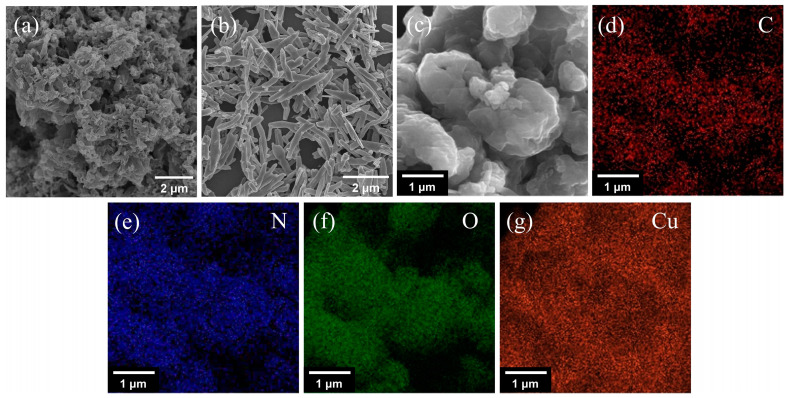
FESEM images of (**a**) g-C_3_N_4_, (**b**) CuO NSs, (**c**) CuO/g-C_3_N_4_, and (**d**–**g**) elemental mapping results of C (red), N (blue), O (green), and Cu (orange) of CuO/g-C_3_N_4_.

## Data Availability

Not applicable.

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
