# Peer review of "Highly Selective Electrochemical CO2 Reduction to C2 Products on a g-C3N4-Supported Copper-Based Catalyst"

_ijms, 2022, doi:10.3390/ijms232214381_

Round 1

Reviewer 1 Report

authors need to provide the FTIR analysis results of all sample and discuss the associated bands. 

please provide the tafel plots of the all catalysts.

cycling stability of the samples must be tested and discuss in revision, 

please refer and cite these relevant articles : 

Journal of Environmental Chemical Engineering

Volume 9, Issue 1, February 2021, 104631;  Journal of Solid State Chemistry 262, 106-11; ACS Catal. 2014, 4, 10, 3637–3643

Reviewer 2 Report

Title: Highly selective electrochemical CO2 reduction to C2 products on g-C3N4-supported copper-based catalyst

Article Type: Full length article

Manuscript Number: ijms-2032051-v1

This work presents a method to prepare CuO/g-C3N4 by catalyst by a simple hydrothermal method with calcination. The synergistic effects between CuO and g-C3N4 enhanced were confirmed.

 My recommendation is that the authors carefully consider the below points, revise appropriately.

1. The authors should consider more representative word in the keywords. Such as: the hydrothermal preparation such the advantageous catalyst.

2. Page 2 line 55~56;”Besides, CO2 is poorly soluble in water, thus ……..” My suggestion is that the authors may consider discussing or supporting more solution as well as suggestion to increase the concentration of CO2 in solution in the section of introduction.

3. Page 8 line 259~260; “Then the ink was added drop by drop on the carbon paper …..1×1 cm2.”My question is that whether the homogeneity of catalyst in carbon paper is good enough to influence the experimental data. And how the authors can ensure the data is repeatable.

4. My suggestion is that the authors may consider adding more explanation or superiority for this catalyst than the others in the section of conclusion.
